# Early-Summer Deficit Irrigation Increases the Dry-Matter Content and Enhances the Quality of Ambrosia™ Apples At- and Post-Harvest

## Changwen Lu

Summerland Research and Development Centre, Agriculture and Agri-Food Canada, Summerland, BC V0H 1Z0, Canada; changwen.lu@agr.gc.ca

**Abstract:** Ambrosia™ is an apple that naturally has limited post-harvest quality retention, which is accompanied by relatively low dry-matter content (DMC). This trial was proposed to improve the DMC of this apple by scheduling deficit irrigation (DI) conducted in a semi-arid orchard in the Similkameen Valley (British Columbia, Canada) in 2018 and 2019. Two irrigation regimes were implemented in the orchard: commercial irrigation (CI) and DI, which was defined as irrigation for 2/5 of the timespan of CI. Five irrigation treatments were conducted: 1—adequate irrigation (AI), which used CI for the whole season; 2—early-summer DI (ED), which used DI from 20 June to 20 July; 3—middle-summer DI (MD), which used DI from 20 July to 20 August; 4—late-summer DI (LD), which used DI from 20 August to 10 days before harvest; and 5—double-period DI (DD), which covered the interval of MD and LD. The DI treatments resulted in a significant decrease from AI $-1.0$ to $-1.5$ MPa in stem water potential (SWP), followed by subsequent recovery. Conversely, SWP did not recover, and instead reached a critical low of $-2.5$ MPa under continued deficit conditions (DD). This, in turn, correlated with significant differences in the DMC among the treatments. Specifically, ED resulted in a rapid and sustained increase in DMC throughout the summer. At the time of harvest, ED resulted in a five-fold increase in the proportion of fruit, with greater than 16% DMC and 15% DMC in 2018 and 2019, respectively, compared to AI. DD resulted in similar levels of DMC elevation compared to ED, but also caused irregular maturation and the increased incidence of soft scald disorder in the post-harvest period. MD and LD had variable effects on DMC, and also increased the incidence of soft scald disorder. Consequently, fruit collected from the ED resulted in the best blush color attributes, higher soluble solid content, and a significant improvement in the post-harvest retention of both fruit firmness and acidity. The ED irrigation model would be recommended as a practical way for Ambrosia™ growers in semi-arid regions to decrease water usage, and to ensure high fruit quality for superior marketing and sustainable production.

**Keywords:** deficit irrigation; water stress; dry matter content; soft scald; quality retention

## 1. Introduction

The Ambrosia™ apple (*Malus × domestica* Borkh.) is the most promoted and rapidly growing cultivar in Canada, and is becoming popular on the global fruit market [1]. The apple is easily recognized for its unique sweetness [2], and is categorized as "a sweet, juicy and flavourful eating apple" in the fresh apple market (Pomiferous; website at https://pomiferous.com/applebyname/ambrosia-id-176, accessed on 21 March 2021). Thus, the crisp and sweet characteristics are key to its marketability. Sensory studies demonstrated that the consumer acceptability of an apple can be further enhanced by improving its taste [2,3]. In contrast, although apple texture is a paramount attribute of quality, optimizations in controlled atmospheric storage have made this less of a forefront issue for the consumer market [4,5]. Customers now favor high taste intensity in fresh market products [3,6]. The total soluble solid contents (SSC) and titratable acidity (TA) play the most important roles in the perception of fruit taste [7], and in creating the flavoring

substances for organoleptic quality similar to those of fresh apples [8]. A main determinant of SSC and TA is the dry-matter content (DMC), which includes carbohydrates, acids and proteins that directly determine taste attributes and flavor profiles [9–11]. Although they are highly correlated with SSC, DMC serves as a better estimate of taste than SSC due to its broad and balanced contents [10–13]. Structural carbohydrates (pectin, fiber, etc.) are used to build and maintain texture and, together with sugars and organic acids, belong to the dry matter in apple fruit. A higher DMC means a higher total carbohydrate level, which translates into better quality and better quality retention [3,7,14]. A post-harvest study showed that apples with better storability had greater DMC accumulation during fruit development compared to apples with lower DMC that rapidly softened post-harvest [15]. In addition, the DMC in pre-harvest, harvest, and post-harvest are highly related [14]. Thus, DMC can be used to predict fruit quality [7,12] and is a new quality metric for apples [3,13].

Ambrosia™ has the unique feature of being ready to eat on the tree with a full fruity taste, which endows this apple with a great advantage on the direct fresh market [16]. However, the Ambrosia™ apple naturally has limited post-harvest quality retention [17,18], which is accompanied by and is synchronous with a relatively low DMC based on our investigation (Table S1). The DMC generally depends on the apple cultivar [14], and abiotic factors such as the weather (precipitation, temperature, wind) and orchard practices affect DMC as well [14]. Apples with a higher DMC (DMC > 16%) lose starch more slowly during storage than those with a low DMC (DM < 13%) [19]. Fruit firmness, both at-harvest and post-harvest, is positively correlated with the fruit DMC [14,19,20]. In addition to inherited and abiotic (weather, location and climatic changes) factors, orchard practices can evidently affect the formation of DMC in apples [14,21–24].

Dry matter accumulates as the fruit grows and matures on the tree [22]. The tree's physiological aspects hold great potential to affect the DMC in apples [22,24]. Orchard irrigation is one of the key factors in the determination of fruit growth and production [16, 23,25]. Deficit irrigation (DI) is increasingly implemented in apple orchards during growth seasons across the world [26,27]. DI involves supplying a tree with an amount of water that is less than 100% of the plant's water needs. DI is a watering strategy that can be applied in different types of irrigation regimes, such as controlled, temporal, continual, or scheduled DI with different levels of water scarcity (mild, regulated, or drought) [26–30]. The correct application of DI requires a thorough understanding of the yield response to water (crop sensitivity to drought stress). The amount of water to be supplied in DI can be calculated or planned based on several parameters, such as the water demand of the plants (the percentage of reference evapotranspiration of the plant), the measurement of plant parameters (e.g., stem water potential), or the water content, moisture, or water holding capacity in the soil. The amount of deficit also depends on the growth stage, and sometimes on the variety or cultivar of a particular species. Studies on irrigation suggest that applying DI during the period after fruit cell division is critical, as fruit growth is slow and shoot growth is rapid during this time. Practical trials have shown that mild water stress applied during this period controlled excessive vegetative growth while maintaining yields [23,26]. The proper timing of DI can not only improve efficiency of water usage but can also benefit fruit production in ways such as preventing the oversizing of fruit growth and enhancing fruit quality [30,31]. A regulated DI trial with Braeburn apples determined that DI applied between 40 and 70 days after full bloom (AFB) (the stage of peak cell expansion) resulted in apples with both the highest marketable yield and the highest red colour density in comparison to adequate irrigation, which was the "commercially irrigated control" [29,32]. There is increased interest in the use of periodic DI on apple trees to improve the fruit quality and enhance the sustainability of orchard production. However, excess or extended water scarcity has made Ambrosia™ apples more susceptible to soft scald disorder after subsequent exposure to chills [1]. There is also a lack of knowledge regarding to the effects of different irrigations on the accumulations of DMC of apple fruit.

The major apple production areas in the northwest coastal regions of North America are semi-arid, and the apple industries in the regions such as the Okanagan–Similkameen

Valley of British Columbia are heavily dependent on irrigation [25]. Practically, commercial irrigation is implemented based on the Tree-Fruit Guide [17], in which apple trees need to be irrigated to full water-holding capacity every 7–10 days during the summer. To date, the effect of deficit irrigation has not been well studied in Ambrosia™ apples. Furthermore, the critical timing of the implementation of DI in this cultivar still lacks detailed management guidelines for growers. This study aims to identify the impact of DI timing on Ambrosia™ apple fruit quality at harvest and post-harvest in a semi-arid region in consecutive years. DMC is the primary outcome measure, and the secondary outcome measures analyzed include the red blush color and compositional attributes.

## 2. Materials and Methods

### 2.1. Orchard and Treatment

The experiment was conducted during the 2018 and 2019 growing seasons using sixteen-year-old apple trees (cv. Ambrosia grafted on M.9 rootstock) grown in a commercial orchard located at lat. 49.16° N and long. −119.74° W in Cawston, British Columbia, Canada. Cawston is located in a semi-arid region within the Similkameen Valley, with hot afternoons, minimal rainfall in the summer (Figure 1), and strong evaporation (the historic weather profile is available on the weather data site at BC/Okanagan South/Cawston EC, at the website https://www.farmwest.com/climate/calculators, accessed on 13 May 2021). The orchard stands on a flat hillside (elevation 454 m) with good ventilation. The soil type is sandy loam with 13% coarse fragments, and is imperfectly drained (BC Soil Information Finder Tool—Province of British Columbia (gov.bc.ca), Sift ID12674). The soil volumetric water content ($VWC$) decline from Day 1 to Day 7 was –0.22 ($\Delta VWC_{\text{Day 7–Day 1}}$ ($m^3/m^3$).

The Ambrosia™ orchard was structured with a super spindle training system, which was situated along a four-wire trellis, spaced 0.45 m apart, with 3.0 m between rows. A total of ninety trees were divided into five blocks of eighteen trees each. Each block (treatment) had three plots (replicates) of six trees, and was surrounded by three guard trees. Each tree yielded 40 to 45 apples in 2018 and 2019 after proper pruning and thinning. Summer trees formed a thin fruiting wall with short branches, and the new shoots were shorter than 0.3 m in length. Maxijet® sprinklers (blue) with 39.6 L/h were employed for irrigation covering a 3-m wetting diameter.

Two irrigation regimes were implemented in the orchard: commercial irrigation (CI) and DI. Based on local industrial regime, CI was defined by calculations in the BC Tree Fruit Production Guide—Irrigation management guide [33], and was implemented here as 10 h of irrigation occurred between 10 pm and 8 am every 10 days approximately, while the DI treatment in this trial was designed as 4 h of irrigation with the same irrigation running for CI. The amounts of water supplements and the soil wet depth were 426 $m^3$/hectare per irrigation and 0.3 m in CI, and 174 $m^3$/hectare per irrigation and less than 0.15 m in DI, respectively.

The trial began 10 days AFB, and five treatments were applied as illustrated in Table 1: 1—adequate irrigation (AI), 2—early-summer DI (ED), 3—middle-summer DI (MD), 4—late-summer DI (LD), and 5—double-period DI (DD).

**Table 1.** Scheme of the regulated irrigation trials conducted in a conventional orchard in 2018 and 2019.

| AI | CI (10 d AFB *–10 d to harvest) | | | |
|---|---|---|---|---|
| ED | CI (10 d AFB–47 d AFB) | DI (47 d–77 d AFB) | CI (77 d AFB–10 d to harvest) | |
| MD | CI (10 d–77 d AFB) | | DI (77 d–108 d AFB) | CI (108 d AFB–10 d to harvest) |
| LD | CI (10 d–108 d AFB) | | | DI (108 d AFB–10 d to harvest) |
| DD | CI (10 d–77 d AFB) | | DI (77 d AFB–10 d to harvest) | |

\* Abbreviations in the table—d: days; AFB: after full bloom; CI: commercial irrigation; DI: deficit irrigation; AI: adequate irrigation with CI for the whole growth season; ED: early-summer DI; MD: middle-summer DI; LD: later-summer DI; DD: double-period DI, which covers the periods MD and LD. The number of days AFB for harvest were 137 in 2018 and 145 in 2019, respectively.

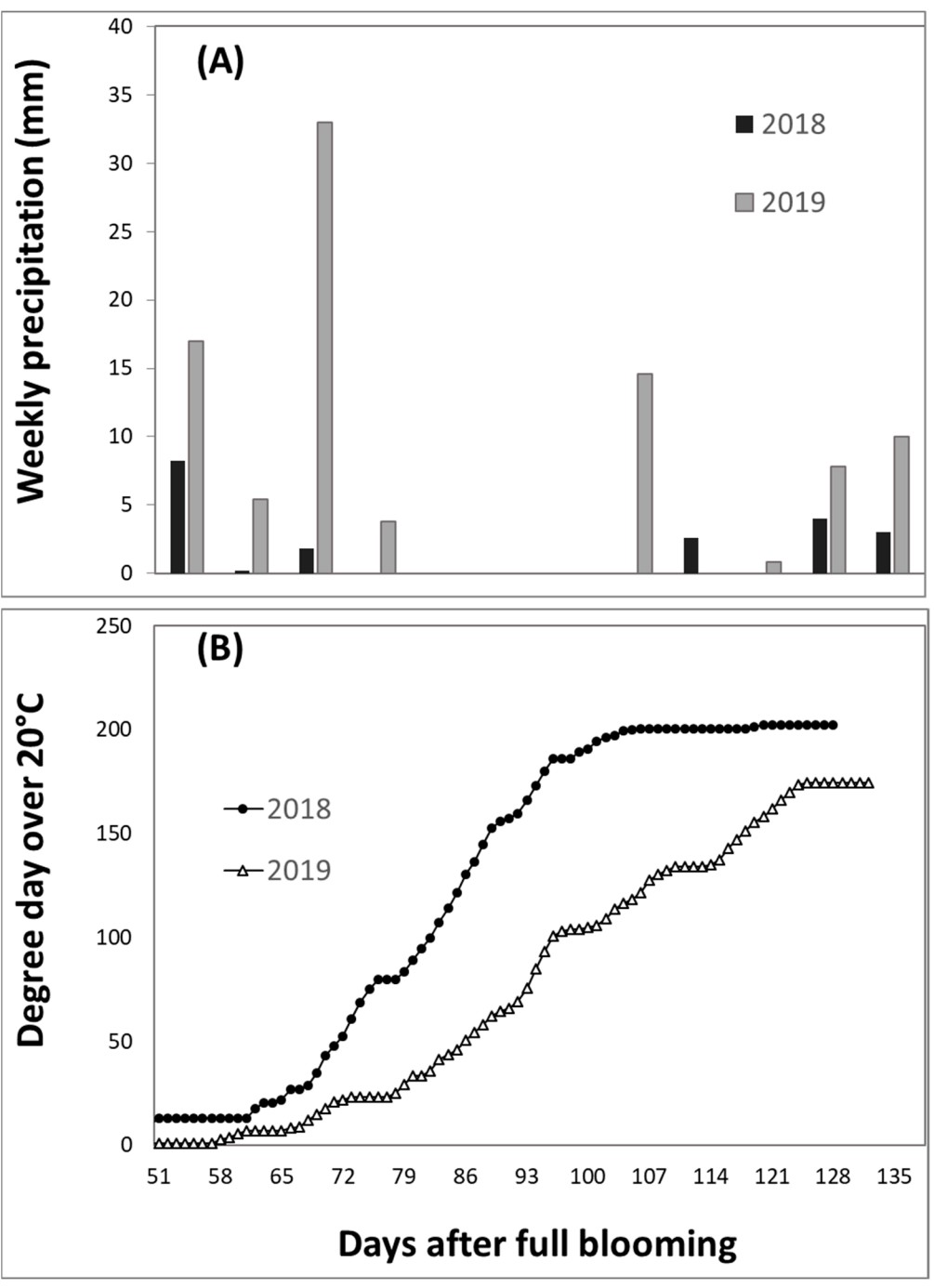

**Figure 1.** Weekly precipitation (**A**) and accumulated degree days above 20 °C that were calculated from the daily mean temperature (**B**) from early summer to harvest in 2 years in Cawston, BC (data obtained for Cawston, BC at Farmwest Historical Weather data site https://farmwest.com/climate/calculators, accessed on 13 May 2021).

### 2.2. Climate and Sampling Timeline Data

The weekly precipitation and accumulated degree days over 20 °C were calculated from historical weather data from June to October in Cawston, both for 2018 and 2019 (data obtained at the "farmwest.com (accessed on 13 May 2021)" weather data site found at BC/Okanagan South/Cawston EC at website https://www.farmwest.com/climate/calculators (accessed on: 12 May 2021)). The fruit expansion rates and fruit quality attributes were highly affected by the accumulated temperature degree days above 20 °C [34], which were calculated from the daily mean temperature data. The full-bloom dates for Ambrosia™

apples in 2018 and 2019 were provided by the Superior Fruit Farm, Cawston, British Columbia. These dates were used to count the days AFB, and were used to set the timelines for starting treatments.

*2.3. Measurements*

2.3.1. Tree Water Relations

The stem water potential (SWP) was measured at midday between 12:30 a.m. and 2 p.m. using a Scholander pressure chamber PMS 1505D (PMS Instrument Company, Al-bany, OR, USA) to evaluate the tree water deficit status [35]. On each sample tree, one short branch with representative, fully expanded and sunlit leaves was enclosed in an equilibration bag (a foil-laminated bag) for 10 min prior to the measurement [36]. Then, the leaves were detached from the shoot, and the stem water potential was determined immediately in the field. The sequence of the measurements was randomized amongst the trees [36].

2.3.2. Prediction of the Fruit Development and Quality Metric on the Trees

Starting in late August, the fruit maturation was assessed weekly using the $I_{AD}$ index measured by a DA meter (Sinteleia, Bologna, Italy), following the protocol modified for Ambrosia™ apples [37]. The DA meter measures the apple nondestructively and gives an index of absorbance difference ($I_{AD}$), which is calculated by subtracting the absorbance at 720 nm from the absorbance at 670 nm. The measurement of a fruit's chlorophyll index gives an indication of the maturation and ripeness state [38]. A single measurement was taken at the transition zone between the sunlit side and the shaded side on each apple [37]. The $I_{AD}$ levels were measured weekly on 20 randomly chosen fruits in each of the three replicates (60 fruits per treatment) in order to predict the timing of the harvest. All of the fruits were chosen from the middle zone of the wall (tree canopy) for minimum variation.

2.3.3. Measurement of the DMC

Starting at the beginning of each treatment during the growing season until harvest, the fruit DMC was measured bi-weekly using a Felix F-750 Produce Quality Meter (Felix Instruments, Camas, WA, USA), which is a handheld visible–near infrared spectrometer that can predict fruit DMC levels non-destructively [14]. The measurement used a predictive model with high linearity ($R^2 = 0.94$) to estimate the DMC of apples that was developed using 100 distinct genotypes from the Summerland Research and Development Centre Apple Breeding Program [14]. The fruit on the tree were measured on the sunlit side, with the lens of the meter tightly placed against the fruit surface. Three replications of 20 fruits per treatment were measured randomly on each day of the investigation (totaling 300 fruits per day). Two days prior to harvest, the frequency distribution of the predicted DMC (FDMC) on-tree was investigated and categorized using the method for apples [14]. The investigation for FDMC was conducted by mapping whole fruits in the experiment field (over 200 in each of the three replications) in all five treatments in both years.

2.3.4. Color Attributes

Red blush coverage on the surface (over-colour) of fruit was assessed visually, and was presented as a percentage. The foreground color (red-blushed over-colour) was recorded as CIE *L*a*b* with a Minolta colorimeter (Minolta CR-300 Chroma meter, Konica Minolta, Tokyo, Japan). *L* is the lightness coefficient, which ranges from 0 (black) to 100 (white); *a* is the major red Chroma coefficient, which represents red when *a* > 0 and green when *a* < 0; *b* > 0 represents yellow, and *b* < 0 represents blue. A blush colour index (to quantify the intensity of the red blush colour for the bicolored apple) was calculated with the following equation for Ambrosia™ apples [1]:

$$BCI = \frac{2000\,a}{L(a^2 + b^2)^{\frac{1}{2}}}$$

where *BCI* is the blush colour index [1], "L" is the measured *L** value, "*a*" is the measured *a** value, and "*b*" is the measured *b** value. In total, 16 apples from each of the tree replicates per treatment were measured.

## 2.4. Harvest Assessment and Post-Harvest Handling

The fruit samples were harvested when the $I_{DA}$ index decreased to 0.5 in 70% of the fruit in the field. For the fruit quality assessment at harvest, 16 apples from each replicate were used for the measurement of the blush color profiles. Then, the apples were further used for compositional analysis. For post-harvest handling, a total of 30 batches of samples including 5 treatments of 3 replications were prepared for the post-harvest test. Each batch contained 25 apples, and two batches representing a replicate were fitted in a storage box with a divider. All of the boxes were placed in air at 0.5 °C for 4 months (commercial period) in order to assess quality retention and the incidence of soft scald disorder. At the end of storage, the fruit were moved to 20 °C for 7 days to ripen before being evaluated.

### 2.4.1. Measurement of the Compositional Properties

The flesh firmness (FF), titratable acid (TA) and soluble solid content (SSC) of the fruit were measured at harvest and after 4 months of air storage at 0.5 °C. Fruit stored with air conditioned to 0.5 °C for 4 months were moved to 20 °C for 7 d to ripen, and were then provided for compositional evaluations. The FF was measured with an 11mm-diameter plunger mounted on an Instrument of Fruit Texture Analyzer (FF) (Model GS-14, Güss Manufacturing (Pty) Ltd., Strand, South Africa). Two punches were made on opposite sides of each apple in the sun/shade transition zone. The plunger was punched into the flesh with a crosshead speed of 200 mm/min, to a depth of 8mm. Two measurements were recorded per fruit, on two peeled areas on opposite sides of the equatorial region of the apple fruit. The FF values were recorded in units of Newtons (N). After the firmness was determined, the apple was then sectioned using a hand-operated Food Prep bench-top corer and wedger (Dito Dean, Rocklin, CA, USA) fitted with an eight-slice wedging and coring head. In total, 10 apples were randomly taken from a replicate, and one wedge taken from each of them was combined and juiced using a Champion Juicer (Plastaket Manufacturing Co., Lodi, CA, USA). In total, 15 mL of the clear juice was diluted to 60 mL with distilled/deionized water. This solution was then titrated with 0.1 N NaOH solution using an automated titrator (Model 719S, Titrino-Metrohm, Brinkmann, Mississauga, ON, Canada) and reported as mg/L malic acid. With apple juice from the same extract for the TA test, the SSC was determined using the Refracto 30PX refractometer (Mettler Toledo, Columbus, OH, USA) and reported in Brix (% *w/v*).

### 2.4.2. Incidence of Soft Scald Disorder

Soft scald (SS) disorders were investigated after 4 months of cold-air storage. The incidence of SS was assessed as 0 (absent) or 1 (present), and then calculated as a percentage of fruits showing the disorder, while the severity of SS was evaluated on a scale of 0 (none), 1 (slight), 2 (moderate), or 3 (severe and unmarketable). The severity of SS was presented as a mean score, which was calculated with the following formula:

$\sum_n$ score of each fruit/total number of fruit.

## 2.5. Statistical Analyses

All of the data with triplicate replicates were subjected to analysis of variance using the SAS GLM procedure (SAS statistical package version 9.3, SAS Institute Inc., Cary, NC, USA). The changes of DA and MDC during the fruit growth on the trees were analysed using Duncan's Multiple Range Test (DMRT); the LS means and pairwise significant letters are shown as dashed lines on the graphs. All of the data on quality attributes at- and post-harvest were used with the least significant difference (LSD) multiple range test to analyze the significance between the five treatments at the *p* < 0.05 level. Tukey's post-hoc test was also conducted for the items above (Supplemental Information Table S3). The statistical

significance between irrigation treatments of stem water potential was analyzed using an ANOVA single-factor model ($p \leq 0.05$, Tukey's Test). The DMC levels near harvest were mapped for the fruit on the trees in all of the treatments, and the FREQUENCY function was used to count the number of test scores that fell within catalogues, which were then further presented as a frequency distribution of predicted DMC categories [14].

## 3. Results

This study describes the significant and quantifiable effects of different irrigation treatments on Ambrosia[TM] apples.

### 3.1. Fruit Development and Maturation

The fruit development was estimated based on the weight at harvest. ED resulted in no significant difference in fruit weight compared to AI, where the fruit weight ranged from 234 to 242 g per fruit in 2018, and from ~245 to 255 g per fruit in 2019 (Table 2). MD resulted in a moderate reduction of the fruit weight in 2018, but showed less impact on the fruit weight in 2019. LD moderately reduced the fruit weight in both years (Table 2). DD significantly reduced the fruit weight by ~45% in 2018 and ~15% in 2019 compared to that of AI. In addition, these smaller fruit were developmentally delayed and not ideally matured at harvest compared to the fruit obtained from the AI treatment.

**Table 2.** Effects of regulated irrigation reductions on the blush color profiles and fruit weight of Ambrosia[TM] apples at harvest in different years.

| Attribute | Year | Irrigation Treatments | | | | |
| --- | --- | --- | --- | --- | --- | --- |
| | | AI | ED | MD | LD | DD |
| a* | 2018 | 42.2 a [1] | 40.2 a | 40.0 a | 30.8 b | 28.1 c |
| | 2019 | 36.9 b | 39.6 a | 37.0 b | 35.2 b | 30.9 c |
| | 2018 × 2019 | 39.6 a | 39.9 a | 38.5 a | 31.6 b | 30.9 b |
| | Year × Treatment | | | $p < 0.0001$ | | |
| BCI | 2018 | 44.2 a | 42.7 a | 41.5 a | 33.8 b | 28.0 c |
| | 2019 | 38.2 ab | 41.2 a | 38.0 ab | 35.4 b | 31.4 c |
| | 2018 × 2019 | 41.2 a | 41.9 a | 39.7 a | 32.6 b | 31.7 b |
| | Year × Treatment | | | $p = 0.0003$ | | |
| Red blush coverage (%) | 2018 | 67.9 a | 66.7 a | 66.0 a | 55.4 b | 38.2 c |
| | 2019 | 61.1 b | 71.7 a | 58.0 b | 58.8 b | 52.2 c |
| | 2018 × 2019 | 64.5 b | 69.2 a | 62.0 b | 57.1 c | 45.2 d |
| | Year × Treatment | | | $p < 0.0001$ | | |
| Fruit weight (g) | 2018 | 242.2 a | 234.7 ab | 222.7 bc | 219.3 c | 153.4 d |
| | 2019 | 255.3 a | 245.1 a | 243.1 ab | 230.6 bc | 221.2 c |
| | 2018 × 2019 | 248.8 a | 238.9 ab | 233.9 bc | 224.9 c | 187.3 d |
| | Year × Treatment | | | $p < 0.0001$ | | |

[1] Mean values with different letters within rows are significantly different at $p \leq 0.05$, as determined by Fisher's protected *t* test using the Proc Mixed models procedure of SAS. The a– star (a*) means Red/Green Value in the color space. The abbreviation letters refer to five treatments of irrigations: AI—adequate irrigation; ED—early-summer deficit irrigation (DI); MD—middle-summer DI; LD—late-summer DI; DD—double-period DI, which covered MD and LD.

The fruit maturations were highly impacted by the irrigation treatments. ED caused a rapid decline in the $I_{AD}$ Index of Ambrosia[TM] fruits in the late growth season in both years (Figure 2). In 2018, the $I_{AD}$ values collected on 13 September from ED were almost equal to the values collected from AI on 18 September (Figure 2A). In 2019, the $I_{AD}$ values collected on 23 September from ED were 0.2 points lower than those from AI (Figure 2B). Linear estimation based on the $I_{AD}$ decline ratio indicates that the fruit from ED were

approximately 10 days ahead of harvest compared to fruit from AI [38]. The maturation process of fruit leveraged by DD was quite different between 2018 and 2019, showing delay or difficulty in maturation in the first year (Figure 2A), in contrast to the early approach to the $I_{AD}$ level of harvest in the second year (Figure 2B). Overall, MD and LD led to minimal changes in the $I_{AD}$ level compared to the AI and ED sites in the 2 years (Figure 2). The investigation of the fruit yield between treatments suggested that one month of DI (ED, MD and LD) did not affect much of the crop under commercial orchard management, except for DD (Table S2).

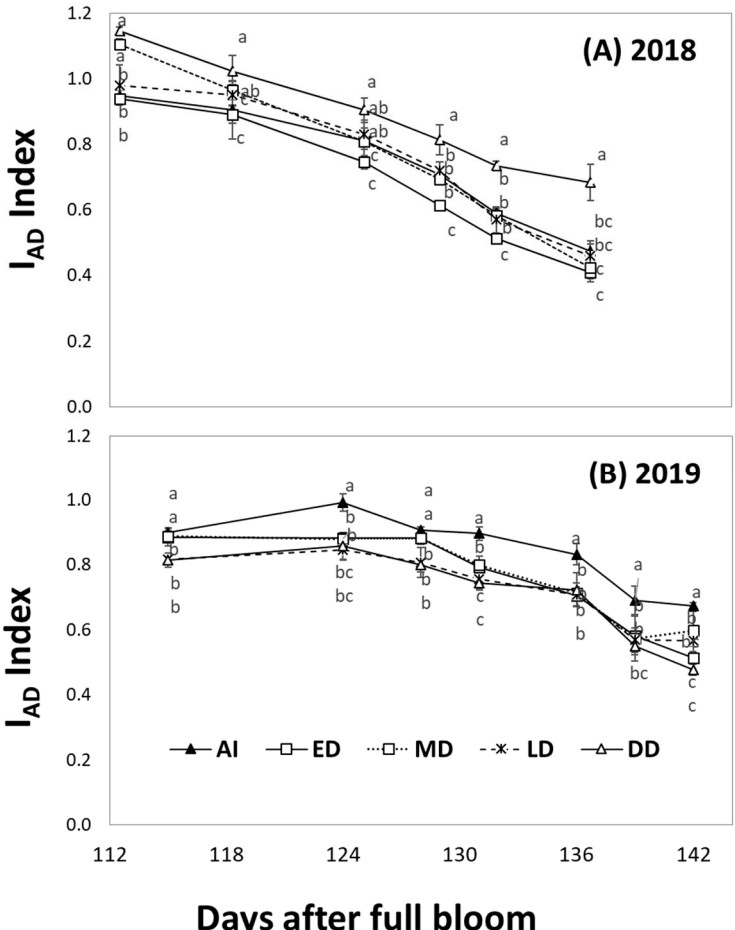

**Figure 2.** Effects of different irrigations on the loss of chlorophyll in fruit skin measured weekly as the $I_{AD}$ Index of Ambrosia™ apples in a conventional orchard in 2018 (**A**) and 2019 (**B**), respectively. The points are the LS means ± standard error of the mean. Means with different letters within a date are pairwise significant ($p \leq 0.05$) differences. AI = adequate irrigation; ED = early-summer deficit irrigation (DI); MD = middle-summer DI; LD = late-summer DI; DD = double-period DI.

*3.2. Changes in DMC*

Data from irrigation experiments conducted in 2018 and 2019 demonstrated that DMC levels are impacted by irrigation (Figure 3): AI maintained a low level of DMC throughout the whole season of fruit growth. ED resulted in an early, rapid increase of DMC that was sustained regardless of the subsequent CI supplements. MD resulted in a 10% relative increase in DMC compared to AI. Significant changes were not observed in LD in comparison to AI. DD resulted in a 20% relative increase in DMC compared to AI (Figure 3). However, the apples from the DD treatment were subsequently more susceptible to soft scald post-harvest (shown details in Section 3.5).

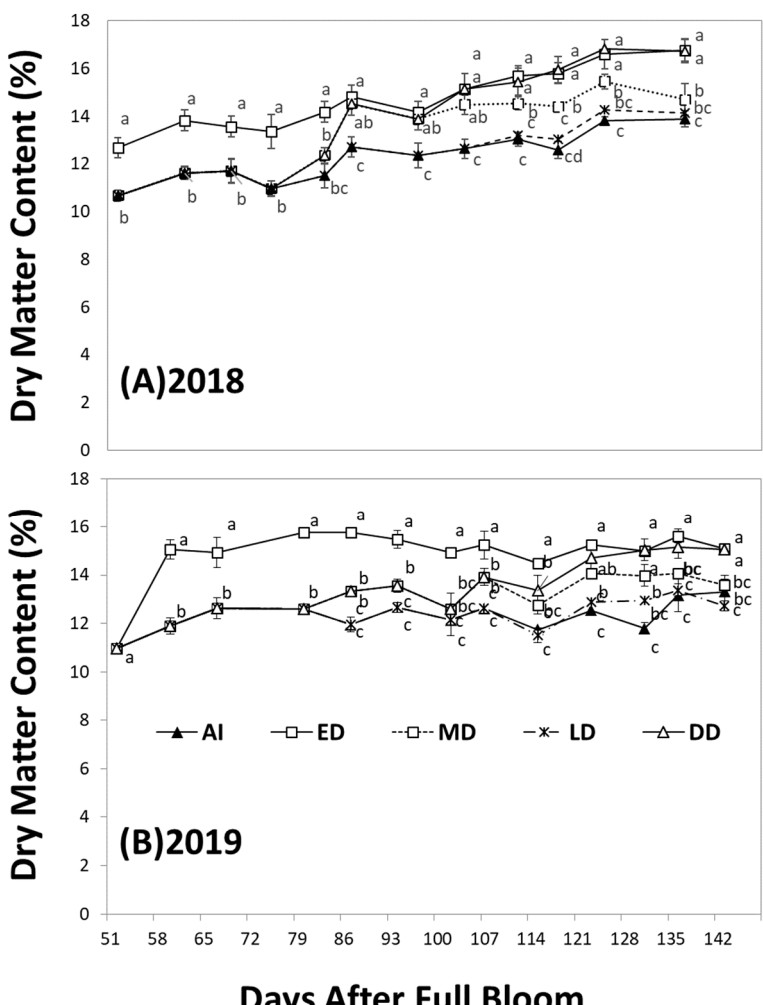

**Figure 3.** Effect of different irrigations on the seasonal changes of the DMC follow-up treatment in Ambrosia™ apples measured with an F-750 NIR spectrometric device in a conventional orchard in 2018 (**A**) and 2019 (**B**), respectively. The points are LS means ± the standard error of the mean. Means with different letters in a date are pairwise significant ($p \leq 0.05$) differences. AI = adequate irrigation; ED = early-summer deficit irrigation (DI); MD = middle-summer DI; LD = late-summer DI; DD = double-period DI.

Frequency analysis further showed that the principal DMC percentage category was 13–14% following AI and LD, up to 15% following MD, and up to 16% following ED and DI. "Whole-field" analysis showed that, at the time of harvest, ED resulted in a five-fold increase in the proportion of fruit with greater than 16% DMC and 15% DMC in 2018 and 2019, respectively, compared to AI (Figure 4). MD also resulted in a slight increase, while LD did not result in significant change compare to AI. DD resulted in similar levels of DMC elevation when compared to ED (Figure 4).

The data on the frequency distribution of DMC categories showed that there were variations in the DMC profiles between 2018 and 2019 (Figure 4). Overall, the DMC of fruit grown in 2018 was higher than that of those grown in 2019.

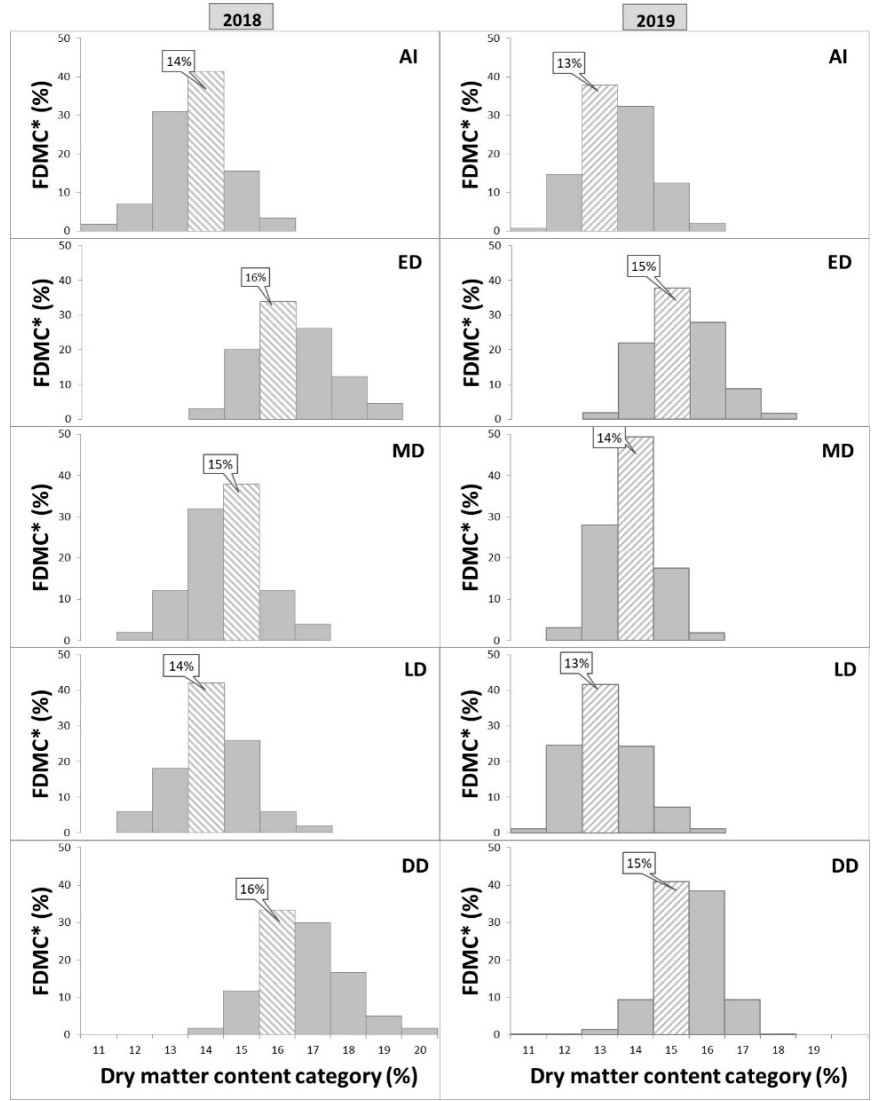

**Figure 4.** Frequency distribution of the predicted dry-matter content (FDMC*) categories of Ambrosia™ apples near harvest from a commercial orchard in Cawston, BC, Canada. The left panel shows the category of 2018. The right panel shows the category of 2019. FDMC: frequency in dry-matter categories, as illustrated by the apple DMC analysis [14]. AI = adequate irrigation; ED = early-summer deficit irrigation (DI); MD = middle-summer DI; LD = late-summer DI; DD = double-period DI. The number within a text box on top of the highest column in each panel indicates the principal (or highest) category of DMC, which was determined with analysis using the SAS General Linear Models procedure ($p \leq 0.05$).

### 3.3. Red Blush Colour on the Surface

The red-blush coverage (RBC) on the fruit surface and its intensity showed that ED led to the highest value of SCI and RBC in both 2018 and 2019. The colour characteristics following AI and MD were similar to those of ED in 2018, but lower than those of ED in 2019 (Table 2). The surface blush colors were lower following LD and DD compared to ED in 2018. Specifically, fruit in the DD group stayed green up (refer to $I_{AD}$ data) to harvest in 2018, and exhibited a light-red or pink surface colour in 2019.

### 3.4. Changes of the Compositional Quality At- and Post-Harvest

All of the samples were evaluated at harvest and after 4 months of air storage. With the exception of the DD samples, the FF levels of the fruit between treatments were comparable at harvest in both years overall. However, the retention values were quite different after

4 months of storage. In 2018, FF decreased by 14.3%, 6.2%, 11.6% and 10.9% in AI, ED, MD and LD, respectively. In 2019, FF decreased by 33.5%, 28.5%, 30.9% and 34.7% in AI, ED, MD and LD, respectively (Table 3). Though fruit from ED were harvested at advanced maturation (Figure 2), this treatment still attained the highest and most consistent FF retention across all treatments in both years (Table 3). In contrast, the FF retention values of fruit from the DD treatment group varied, with only an 8.8% loss of value in 2018, but more than a 34.7% loss in 2019. The highest FF value from DD in 2018 was associated with "small green" fruit (Table 3) because of its immature stage at harvest (Figure 2). In the comparison of the yearly changes, the FF of fruit grown in 2018 had better retention than those grown in 2019 (Tables 3 and 4). This result was well correlated with the DMC differences (Figures 3 and 4).

**Table 3.** Changes of the compositional quality attributes of Ambrosia[TM] apples between the at- and post-harvest measurements in the years 2018 and 2019.

| Attribute | Year and Stage | | Irrigation Treatments | | | | |
|---|---|---|---|---|---|---|---|
| | | | AI | ED | MD | LD | DD |
| FF (N) | 2018 | at-harvest | 67.7 b [1] | 66.4 b | 67.4 b | 66.1 b | 75.4 a |
| | | post-harvest | 58.0 c | 62.3 b | 59.6 bc | 58.9 bc | 68.8 a |
| | 2019 | at-arvest | 71.0 a | 70.1 ab | 70.6 ab | 69.5 bc | 68.9 c |
| | | post-harvest | 47.2 bc | 50.1 a | 48.8 ab | 48.9 ab | 45.0 c |
| TA (mg malic acid L$^{-1}$) | 2018 | at-harvest | 262.0 ab | 267.2 ab | 261.9 ab | 262.9 ab | 283.9 a |
| | | post-harvest | 187.1 c | 222.6 b | 186.8 c | 185.7 c | 234.4 a |
| | 2019 | at-harvest | 262.3 a | 247.5 b | 259.3 b | 250.1 b | 248.7 b |
| | | post-harvest | 165.8 d | 191.1 a | 172.5 c | 179.8 b | 175.5 bc |
| SSC (%) | 2018 | at-harvest | 13.3 c | 15.5 a | 13.9 b | 13.5 bc | 15.9 a |
| | | post-harvest | 13.4 c | 15.2 a | 13.7 b | 13.7 b | 15.1 a |
| | 2019 | at-harvest | 11.4 c | 13.2 a | 11.5 c | 12.1 b | 13.2 a |
| | | post-harvest | 12.3 d | 14.6 a | 13.3 b | 12.7 c | 14.4 a |

[1] Mean values with different letters within rows are significantly different at $p \leq 0.05$, as determined by Fisher's protected *t* test using the Proc Mixed models procedure of SAS. The abbreviation letters refer to five treatments of irrigations: AI—adequate irrigation; ED—early-summer deficit irrigation (DI); MD—middle-summer DI; LD—late-summer DI; DD—double-period DI, which covered MD and LD.

**Table 4.** Combined statistical significance of the effects of the irrigations on the changes of the compositional attributes of Ambrosia[TM] apples in the years 2018 and 2019.

| | FF (N) | TA (mg Malic Acid L$^{-1}$) | SSC (%) |
|---|---|---|---|
| Year | <0.0001 | <0.0001 | <0.0001 |
| Treat | <0.0001 | <0.0001 | <0.0001 |
| Stage | <0.0001 | <0.0001 | <0.0001 |
| Year × Treat | <0.0001 | <0.0001 | 0.0002 |
| Year × Stage | <0.0001 | 0.0025 | <0.0001 |
| Treat × Stage | <0.0001 | <0.0001 | 0.0034 |
| Year × Treat × Stage | <0.0001 | 0.0027 | <0.0001 |
| 2008 | 65.0 a [1] | 233.7 a | 14.3 a |
| 2009 | 59.0 b | 215.1 b | 12.8 b |

[1] Mean values with different letters in columns are significantly different at $p \leq 0.05$, as determined by Fisher's protected *t* test using the Proc Mixed models procedure of SAS. Years: 2018 and 2019; Stage: at- and post-harvest; Treat = treatment, e.g., five irrigations: adequate irrigation, early-summer deficit irrigation, middle-summer DI, late-summer DI, and double-period DI, which covered MD and LD.

The TA contents of the fruit between treatments was evident at harvest in both years, except for DD. However, the differences of the maintained values after 4 months of storage were significant (Table 3). In 2018, TA decreased by 27%, 15%, 29% and 29% in AI, ED, MD and LD, respectively. In 2019, TA decreased by 36%, 26%, 33% and 30% in AI, ED, MD and

LD, respectively (Table 3). ED resulted in the least degradation and retained the highest value of TA, and this impact was consistent in both years (Tables 3 and 4). The TA from fruit following DD treatment showed a higher retention value in 2018 but a lower retention value in 2019. Overall, the retention values were comparable in the two years; however, in 2018 the retention values were as high as 83%, compared to 2019 with retention values of 70% (Table 3).

The SSC data highlighted the differential impacts from irrigation: compared to AI, ED and DD yielded similar values, which were increased over a relative 16% at harvest and 15% after storage in both years (Table 3). MD and LD resulted in an increased SSC, significantly higher than AI, which was variable and still far below the values from ED and DD (Table 3).

### 3.5. Soft Scald Disorder in Storage

Regarding soft scald (SS) disorder, ED and AI resulted in similar low incidences of SS. Following MD and LD, approximately 25% of the apples had SS in 2018, but there was only minimal SS following these treatments in 2019. DD led to critical SS disorders compared to the other treatments after 4 months of cold-air storage (Table 5), with SS occurring in 63.8% of apples in 2018 and 42.8% of apples in 2019. Notably, the apples from DD showed the SS disorder more critically in 2018 than in 2019, in both incidence and severity (Table 5). In 2018, the symptoms of the disorder in the DD batches began to show within approximately two months of storage, and the incidence of the disorder was delayed to three months of storage in the same circumstances in 2019 (data not show).

**Table 5.** Soft scald disorders of Ambrosia™ apples in 4 months of air storage at 0.5 °C amongst samples from different irrigation treatments in the growth seasons of the years 2018 and 2019.

| Disorders | Year | Irrigation Treatments | | | | |
|---|---|---|---|---|---|---|
| | | AI | ED | MD | LD | DD |
| Incidences (%) | 2018 | 5.8 c [1] | 2.9 c | 24.6 b | 27.5 b | 63.8 a |
| | 2019 | 5.9 b | 4.8 b | 5.2 b | 6.2 b | 42.8 a |
| | 2018 × 2019 | 5.9 c | 3.7 c | 14.9 b | 16.9 b | 53.1 a |
| | Year × Treat | $p < 0.0001$ | | | | |
| Severity (worst = 3) | 2018 | 0.06 c | 0.03 c | 0.33 b | 0.36 b | 1.16 a |
| | 2019 | 0.09 b | 0.06 b | 0.08 b | 0.11 b | 0.62 a |
| | 2018 × 2019 | 0.08 c | 0.04 c | 0.21 b | 0.23 b | 0.89 a |
| | Year × Treat | $p < 0.0001$ | | | | |

[1] Mean values with different letters within rows are significantly different at $p \leq 0.05$, as determined by Fisher's protected *t* test using the SAS GLM procedure. The abbreviation letters refer to five treatments of irrigations: AI—adequate irrigation; ED—early-summer deficit irrigation (DI); MD—middle-summer DI; LD—late-summer DI; DD—double-period DI, which covered MD and LD.

### 3.6. Stem Water Potential

In this study, the stem water potential (SWP) under AI treatment was −1.0 MPa in July (Figure 5), and increased to −0.5 MPa in the late season (Figure 6). Meanwhile, under treatment, DI resulted in a decrease of 1.5 MPa in mid-July (ED in Figure 5), and to the most extreme low of −2.5 MPa in the late season (DD in Figure 6A). The value of −1.5 MPa recorded during ED treatment impacted neither the fruit development (Table 2) nor sustainable production (e.g., fruit small). As is further indicated in Figure 6, the trees that experienced ED were subsequently able to resume a normal SWP value.

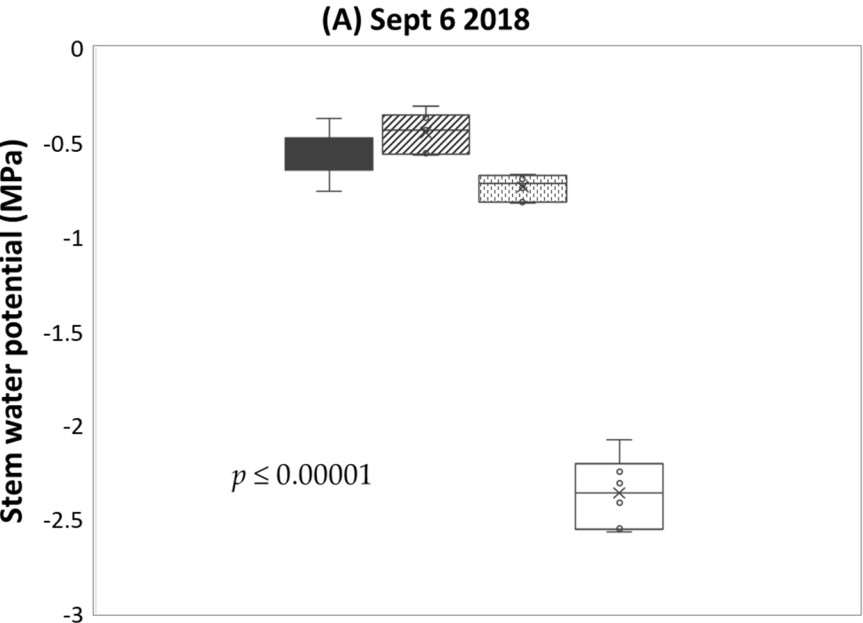

**Figure 5.** Stem water potential in Ambrosia[TM] trees in early summer. The data were collected on July 16 2019 (73 days AFB), with eight replicates. The statistical significance between irrigation treatments was analyzed using an ANOVA single-factor model ($p \le 0.05$). The box shows the minimum, first-quartile, median, third-quartile, and maximum values. The bars with error bars are means $\pm$ standard errors. AI = adequate irrigation; ED = early-summer deficit irrigation (DI); MD = middle-summer DI; LD = late-summer DI; DD = double-period DI.

**Figure 6.** *Cont.*

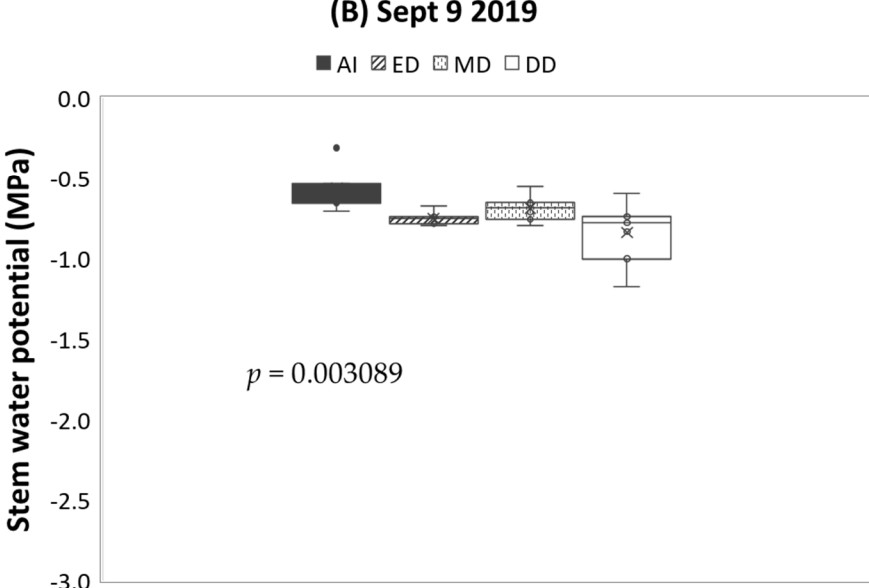

**Figure 6.** Stem water potential in Ambrosia<sup>TM</sup> trees in late summer. The data were collected on 6 September (125 days AFB) 2018 (**A**) and on 9 September (128 days AFB) 2019 (**B**), with eight replicates, respectively. The statistical significance between irrigation treatments was analyzed using an ANOVA single-factor model ($p \leq 0.05$). The box shows the minimum, first-quartile, median, third-quartile, and maximum values. The bars with error bars are means ± standard errors. AI = adequate irrigation; ED = early-summer deficit irrigation (DI); MD = middle-summer DI; DD = double-period DI.

## 4. Discussion and Suggestions

Irrigation is a substantial factor governing tree growth, fruit development and fruit quality [23]. Excessive irrigation causes vigorous growth [16] and the development of diseases in fruit, such as bitter pits [39]; in contrast, water deficit led to the irregular growth of fruit (Tables 2 and 3) or even to the death of the tree (6 out of 18 trees died in DD in 2018, as indicated in Table S2). Because the climate in the Okanagan area was wet in the spring from the winter snow, with major rainfall for the whole growth season in the early summer (Figure 1), an early reduction in irrigation did not cause adverse affects in the trees. That is why the ED practice works out for quality improvement with fewer flaws in the experimental site.

An SWP of −1.5 MPa is considered a general benchmark for plant water deficit, and is the wilting point for many crops [22]. However, this did not seem to be the case in the present study, which suggested that dwarfing Ambrosia<sup>TM</sup> can tolerate early-season short-term DI. As is further indicated in Figure 6, the trees that experienced ED were subsequently able to resume a normal *SWP* value. However, an SWP of −2 MPa seems to be the critical point of triggering DI stress such as "small" and "green" fruit (Tables 2 and S2) and tree damage (there was no crop in the second year; data not show). These preliminary measurements only revealed the basic relation between *SWP* and tree physiology tolerance; accurate *SWP* measurement to indicate tree water demand needs further study. Additionally, in semi-arid regions, Ambrosia<sup>TM</sup> apples seem highly sensitive to the water status of the tree during the late growth season, in which the SWP has to be over −1 MPa, otherwise the fruit would suffer SS disorder in the subsequent post-harvest stage (such as DD in 2019). Similar observations were documented previously in this apple in a row-cover trial in the 4 weeks up to harvest, in which the solid film preventing water penetration to the soil contributed to inducing heat and/or water stress, and caused SS disorder post-harvest [1].

The timing of the water deficit is the key to properly leveraging water management [17,23]. The growth cycle of the fruit tree is classified in five stages: stage 1—budburst and flowering (fruit growth by cell division); stage 2—beginning of rapid shoot growth while fruit grows slowly; stage 3—beginning of fruit filling (rapid fruit growth with cell expansion while the shoots grow slowly); stage 4—harvest; and stage 5—leaf fall [23]. Amongst these, stage 1 occurs before "early summer." Though water supplement is highly demanded during this stage, orchardists in the Okanagan–Similkameen region are not concerned about irrigation; there is naturally plenty of melted snow and rainfall, and temperatures are relatively low in this stage (Farmwest.com, accessed on 21 March 2021). However, stage 2 and stage 3 are highly challenging times for water management due to the dry and hot weather. Conventionally, growers used to supply more water via frequent irrigation or extended irrigation (represented by CI in this study). However, this approach to irrigation can potentially negatively impact either fruit quality (overly large size, irregular shape, plain taste, etc.) or tree vigor. Therefore, the careful management of irrigation based on water demand in these two stages is important. This study demonstrates that the timing of the water deficit increased the DMC in Ambrosia$^{TM}$ apples. How and why does this work? Dry matter mainly represents the carbohydrate levels in fruit [22]. Water deficit in early summer is assumed to lead fruit to acquire a higher share of carbohydrate production via the suppression of vegetative growth [22,23]. ED allows the resumption of CI after the early-summer DI. This enables proper photosynthesis activity and fruit development during the subsequent period of rapid fruit expansion (stage 3) (Table 2). Early studies on apple trees [40] have demonstrated that fruiting spur leaves have two elevated photosynthetic rates correlated with the fruiting process during the growing season: the first period of increased photosynthetic rates was during the bloom period (stage 1 in the growth cycle classified by Boland et al. [23], and the second was the rapid fruit growth from midsummer to harvest (stage 3). This suggests that ED treatment was rendered in a moderate period of photosynthetic activity while fruit development was not heavily impacted. Additionally, the Ambrosia$^{TM}$ tree has a tendency for strong lateral branch development and upright growth [18]; implementing DI in the stage of rapid vegetative growth (stage 2) would be a timely control for vigor growth. A pomology study found that the implementation of water deficit treatment between the 40 and 70 days AFB successfully controlled vegetative growth and produced apples achieving the highest red color density [29]. After the DI regime of ED, the tree SWP attained the same level as AI (Figure 6) and an equal rate of photosynthetic activity (data not shown) to those from AI, suggesting that the treatment retained the physiological ability to ensure fruit which were well balanced in size, quality and reproduction (Tables 3–5). However, water deficit across the whole period of the fruits' rapid growth (stage 3) caused water stress, which raised the DMC value simply through water loss from the fruit [36] (Table 2). This could explain the worse blush color profiles and size, as well as the difficulty in maturation of the fruit in the DD treatment in 2018 under heat and a dry climate (Table 2). Regarding the effect of water deficit stress on increasing DMC, it may be possible that ED enables a physiological impact on fruit in high correlation with all of the quality parameters, but DD attained a higher DMC via a simply mechanical process (such as fruit dehydration and shrinkage), with poor association with quality attributes in consecutive years (Tables 3 and 4). The details of the physiological and cytological mechanism need to be more precisely studied in the future.

This study further confirms the proposition that fruit obtaining a higher DMC have better quality [3,15]. Consistently, the DMC level was highly correlated with the SSC value of fruit both at- and post-harvest in the two years (Table 3 and Figure 4). However, the DMC was not always positively correlated with the blush color profile. In this study, DD acquired a higher DMC in the two years in which the color attributes presented higher values in 2018 but were worse in 2019 (Table 2). This suggests that the coloration of apple fruit is based on physiological activities which are highly water dependent [22,36]. Notably, this study suggests that the DMC level was not highly correlated with the flesh firmness (FF) and acidity recorded at harvest, but is was with their retention after storage (Table 3).



Based on the data collected from the treatments of AI and ED, the correlation analysis indicated that for the fruit DMC with the loss of FF, r = −0.87 in 2018 and −0.86 in 2019; with the loss of TA, r = −0.82 in 2018 and −0.76 in 2019. In comparison to the values at harvest, ED retained an FF value over 94% in 2018 and near 71% in 2019, while FF retention from AI pre-treatment was lower, at 85% in 2018 and 66% in 2019 (Table 3). The retention ratio of TA was 83% and 77% in ED, and 71% and 63% in AI, in 2018 and 2019, respectively (Table 4). The cell wall structures led to differences in the softening rates during the apple (*Malus* × *domestica*) fruit growth [14]. DMC is the primary material of the cell wall [22]. This may be why the ED fruit that possessed a high DMC had a better retention of FF. DMC accumulation seems to be highly associated with seasonal temperature and precipitation (Figure 1); therefore, the DMC accumulation was different between different years. The results obtained in this study highlight that both the mean (Figure 3) and category (Figure 4) fruit DMC values recorded near harvest were about one unit higher in 2018 than in 2019 across the treatments. The percentage unit in the most frequent categories was 16 in year 2018 and 15 in year 2019 (Figure 4). Similar DMC differences between different years have been reported previously, such as a 4% gap in Royal Gala apples [15]. This is not surprising because of two reasons. First, the weather was different between the two years; the accumulated degree days above 20 °C for early- and middle-summer in 2018 was about double that in 2019 (Figure 1B). Horticultural studies [34] have indicated that the mean fruit weight from warm post-bloom treatments is up to four times greater at harvest than that from cool-temperature treatments. Additionally, fruit from warm post-bloom temperature conditions have a higher soluble solids concentration and advanced maturation than fruit from cooler temperatures [34]. On top of the temperature impact, differences in water statuses have a further effect on DMC. Secondly, there was less precipitation, and therefore more drought-like weather, in 2018, which amplified the impact from water stress and further increased the accumulation of DMC. In Figure 3 showing DMC changes across the growth season of the fruit, the plot area delineated by the dashed line suggests that looking at the DMC accumulated over time may be more interesting than the amount of DMC attained at one time. In other words, maintaining DMC at a high level over a long period of time is more desirable than attaining the same level but for a shorter period of time. The method for the estimation of the accumulated DMC may mimic the formula of "degree d" as the sum of the DMC value in each recording × interval (days) between investigations × frequency of investigation in whole season (e.g., $\sum_n$ DMC value × days of interval, where n = the frequency of investigation from fruit set to harvest), which may deserve future study.

The fruit maturations were highly impacted by the irrigation treatments. ED caused a rapid decline in the $I_{AD}$ Index of Ambrosia$^{TM}$ fruits in the late growing season in both years (Figure 2). In 2018, the maturation based on $I_{AD}$ values from ED was estimated to be completed 5 days earlier than that of the fruit from AI (Figure 2A); similarly, in 2019, maturation was estimated to occur about 10 days ahead of harvest in fruit from ED compared to those from AI according to the $I_{AD}$ decline ratio. The maturation progress of fruit treated with DD was quite different between 2018 and 2019, showing delayed maturation in the first year (Figure 2A) but an early approach to the $I_{AD}$ level of harvest in the second year (Figure 2B). This suggests that water deficit to the proper extent, e.g., allowing photosynthetic activity to recover in stage 3, can lead to early fruit maturation; otherwise, extreme or irreversible drought caused delayed or disordered maturation. Ambrosia$^{TM}$ is a relatively late-maturing apple with high susceptibility to chilling injury. The quick drop in night temperatures in the fall may increase the incidence of soft scald and core breakdown (personal communication with Dan Tayler, the apple industry operator in Cawston Cold Storage Ltd., Cawston, BC, Canada, 2018), and the earlier maturation of apples under ED allows an earlier and more flexible harvesting window to reduce or avoid chilling-related disorders [5].

## 5. Conclusions

This study tried to improve the DMC of Ambrosia™ apples using the optimal timing of deficit irrigation conducted in a semi-arid orchard in the Similkameen Valley (British Columbia, Canada) in 2018 and 2019. It demonstrated that the ED model in the growth season results in rapid increasing and sustained higher DMC compared to the DMC in fruit treated with AI throughout the growth season. ED is further able to yield better levels of blush color, SSC, FF, and TA than the other irrigation treatments in this study. ED also has a lower incidence of SS disorder compared to the other DI treatments tested. Thus, this study suggests that the ED irrigation model should be recommended as a practical way for Ambrosia™ growers in semi-arid regions to decrease water usage, and to ensure high fruit quality for superior marketing and sustainable production.

**Supplementary Materials:** The following supporting information can be downloaded at: https://www.mdpi.com/article/10.3390/horticulturae8070571/s1. Table S1: DMC[1] levels and storage period of some common apple cultivars of Canada in cold-air storage; Table S2: Tree survival and fruit yields of Ambrosia™ apples following different irrigations in the years 2018 and 2019; Table S3: Tukey's test outputs.

**Funding:** This study was part of the Agri-Science Program—005 British Columbia Fruit Growers' Association "National Program for the Development of Improved Tree Fruit Varieties Activity 8", funded by the New Tree Fruit Varieties Development Council, and Agriculture and Agri-Food Canada (J-002066).

**Institutional Review Board Statement:** Not applicable.

**Informed Consent Statement:** Not applicable.

**Data Availability Statement:** The data are archived in the Tree Physiology lab, AAFC's Summerland Research and Development Centre, and are available upon request.

**Acknowledgments:** The author wishes to acknowledge the funding provided by Agriculture and Agri-Food Canada and the New Tree Fruit Varieties Development Council, and sincerely appreciates Hao Xu for valuable contributions to the investigations of tree–water relations and leaf photosynthesis. In particular, the author is grateful to Peter Toivonen for sharing insights on fruit dry-matter research and partial supplements, and to Similkameen Superior Fruits Ltd. in Cawston, British Columbia, for providing the study materials and technical assistance. The author also thanks Brenda Lannard, Danielle Ediger, and the extensive number of co-op students 2018–2019 for their technical support.

**Conflicts of Interest:** The author declares no conflict of interest.

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
