# Peer review of "Early-Summer Deficit Irrigation Increases the Dry-Matter Content and Enhances the Quality of Ambrosia™ Apples At- and Post-Harvest"

_horticulturae, doi:10.3390/horticulturae8070571_

Round 1

Reviewer 1 Report

The manuscript investigated the effect of deficit irrigation applied in different fruit development periods on dry matter content and quality attributes of Ambrosia apples. The study is interesting and useful to growers, however there are several issues that need to be addressed before publication. First of all, 2 years of data may not be sufficient if Ambrosia is an alternate bearing cultivar (please specify). In addition, the experimental design must be clarified and the statistical analysis must be re-done using the appropriate tests. The introduction needs some editing and some more appropriate studies should be cited. The manuscript needs also to be carefully checked for some minor language issues. These and several other comments are marked in the attached file and must be properly addressed before accepting the manuscript for publication.

Reviewer 2 Report

The manuscript entitled "Early summer deficit irrigation increases dry matter content and enhances quality of Ambrosia™ apple at- and post-harvest" by Changwen Lu, provides very useful information regarding the postharvest quality of Ambrosia (TM) apples. 
The manuscript is very well written. The introduction section provides the appropriate information to the reader. The Material and Methods section is very descriptive. The results are very well presented and the Tables and Graphs are well organized. In the discussion section, all the necessary information is given to support the results.
The beneficial effect of early deficit irrigation is documented and is well proposed as a potential cultivation technique for Ambrosia apples, to increase the dry matter content and their nutritional value.
The manuscript is worthy of being published in its current form.

Reviewer 3 Report

The main aim of the paper is to assess the influences of deficit irrigation in dry matter content of Ambrosia apples.

The study is interesting and gives interesting information. But some information is needed to clarify the experiment.

Specific comments

Line 60. What about a determination of the individual sugars and acids content? It would be a better approach? And nutritional content? DMC is a general value that can include a lot of compounds. It is an indirect index but very rough compared to other more specific techniques.

Line 64. Explain why that happens. What type of conservation affects?

Line 66. This is a result? So, it should no be in the introduction.

Line 98. References and information about DI in other fruit trees can be given.

Line 130. Five blocks and five treatments. Each block has three plots but all with the same treatment? Or different treatments within the block? How many trees per treatment?

Line 150. The experiment was done in 2018 and repeated in 2019? The irrigation dates were the same both years?

Line 207. The other of the figures should follow the order in the text. Figure 3 is explained before figure 2 for example.

Line 271 This data refers to both years?

Line 325. Mean values should be with their error. Each abbreviation should be explained (BCI for example).

Line 363. In Table 4. In FF attribute is riten “at-arvest “ in stead of “at harvest”

Line 370. Table 5 is year 2008 and 2009? Should be 2018 and 2019? This table gives little information. There are statistical differences as observed in previous tables.

Line 412. Dead trees only in 2018. Why not in 2019?

Line 439. Why is missing LD treatment in this figures?

Line 492. Higher quality? Due to higher SSC and TA? Or other factors?

Line 560. Repetitions in other locations will give the same result? Discuss

Line  562. Figure S1 and Video S1 are no available.

Round 2

Reviewer 1 Report

After the first round of review, the manuscript has been significantly improved. However, there is still a major concern that needs to be addressed properly. If Duncan's multiple comparison test has been (erroneously) used in several papers in the literature, that does not mean that it is ok to keep using it. The Authors should reanalyze their data using Tukey's multiple comparison instead for a proper interpretation of results before publishing the manuscript. A couple of other comments are marked in the attached file.

Author Response

Thank you for your time to review my manuscript with professional comments. Please see my response file as attached. 

Reviewer 3 Report

Improvements have been made and it is suitable for its publication

Author Response

Thank you very much for your possitively appraisal on my manuscript! 

This manuscript is a resubmission of an earlier submission. The following is a list of the peer review reports and author responses from that submission.

Round 1

Reviewer 1 Report

The article describes how water management affects storability - particularly the dry matter content (DMC) of apples.

Interesting, relevant topic, but the description (or lack thereof) of irrigation regime is very confusing.

Deficit irrigation involves supplying plants with an amount of water that is less than 100% of the plant's (crop, tree) water needs. The amount of water to be supplied in deficit irrigation can be calculated or planned based on several parameters, such as the water demand of the plants (percentage of reference evapotranspiration of the plant), measurement of plant parameters (e.g. SWP measured by the author) or the water content/potential in the soil. The amount of deficit also depends on the growth stage, and sometimes on the variety or cultivar of a particular species.

The part where the article should be substantially improved is the description of water regime treatments, because none of the water regime treatments author refers to are explained at all.

Even in the introduction, L72 - L84 are confusing.

What is the difference between reduced irrigation and deficit irrigation? Use deficit irrigation to avoid confusion. The author uses phrases like conventional irrigation, commercial irrigation - conventional management (supplementary materials).

Adequate irrigation is what?

L97 soil moisture can be dried to 200 kPa - you are referring to soil water tension? Did you measure soil water status? Why did you make this reference as soil water status is not mentioned anywhere else in the article?

L117 10.5 GPH is that gallons per hour? Improve to liters (l/h) or better to mm.

In Materials and Methods L117 - L129 the description of water balance (amount, timing, duration) is inadequate and needs to be improved.

Minor comments - write abbreviations in the table headings (e.g. T S1 - DMC).

L71 tree fruit guidance- should there be guidelines here?

L71 Irrigation is only one measure - a stable water supply appropriate to the growth stage is important

Reviewer 2 Report

In the submitted manuscript by Changwen Lu entitled “Early summer reduced irrigation increases dry matter content and enhances quality of Ambrosia™ apple at- and post-harvest”, the author examined the effect of reduced irrigation (RI) at different apple’s fruit stages via determination of dry matter content (DMC), the yield of high DMC fruits and other fruit physiological traits such as soluble solid content, firmness, acidity, and colour. The author found that early-summer RI led to a rapid increase of DMC. Finally, the author recommends ER irrigation model in semi-arid regions for Ambrosia™ growers to minimize water usage and to produce high apple fruit quality based on DMC. Overall, the manuscript is well-written, and generally, the author's data and statical analysis are clear. The aim and scope of the Horticulturae journal are in line with the current manuscript. However, there are some issues that should be carefully addressed.

Major and minor comments,

  • Lines 105-106: The author should provide a soil analysis regarding particle size analysis (sand, silt and clay content) and define bibliography or experimental soil water holding capacity.
  • Lines 118-120: The author should explain in more detail what does that means (10 hours or 4 hours of irrigation)? The author should provide for instance 1. the depth of irrigation in the soil in both conditions, 2. the mm of the irrigated water per hectare, 3. the soil moisture at 15 mm of soil depth across the time. In general, the author should incorporate data about the irrigation process concerning the soil.
  • The quality of the figures should be improved.
  • The author should construct and provide a final / schematic figure to summarize the current results. For example, point out the beneficial effect of reduced irrigation at the early-summer in dry matter content of Ambrosia™ apple compared to other reduced irrigation time points and commercial irrigation.

Reviewer 3 Report

This paper is focused on issues affecting the apple cv., Ambrosia, which is a relatively new entry on the global apple market, and still lacks detailed management guidelines for growers. The merit of the paper is to apply some ecophysiological measurements to a number of fruit quality parameters, to try and determine connections between pre-harvest conditions and post-harvest quality, including occurrence of storage disorders.

The amount of work, the fact that 2 years of observations have been conducted, are positives of the work. However, I cannot say I feel that the amount of data collected, its presentation and interpretation provide innovative knowledge, that warrants publication in the present form. RDI was proposed since its beginnings as a way to regulate vegetative growth without negative consequences on fruit growth. Notably, it came out of Tatura, Victoria, a place that can have extremely warm and dry conditions during spring, compared to other parts of the World, which make water retention extremely powerful in controlling vegetative growth. There is not much new here. Since the climate in the experimental site is indicated as wet in the Spring, in addition, an early reduction in irrigation may not cause too much adverse affects in the trees. I may have missed it, but I have not seen this possibility discussed.

While many experimental approaches are quite state of the art, some items in the literature are old (40, 4, 34, 30) and could have been substituted by more recent ones.

One difficulty in assessing the paper lies in the fact that no phenology is given, so placing the whole experiment in context is difficult. I may guess that flowering happens much later here than in other parts of the world, and probably growth is fairly fast, based on fruit size at harvest. However, not knowing the date of full bloom makes it very hard to understand whether cell division could still have occurred or not when the ED was applied. Warrington et al. (34), for example indicated that low temps during the first 40 DAFB caused smaller fruit size although, since they did not count cell numbers, they could not prove that was the cause. Further work from Cornell, and some from my own lab, both published and unpublished, have shown that, the colder the Spring, the longer cell division would last, although the total number of cells per fruit might not be very different. I give this long account of just one factor that I think could have a deep effect on the results (the 2 years are quite different from the point of view of degree day accumulation) and that I did not see discussed properly.

I also miss any reference, in the discussion, to vascular flows underpinning phloem and xylem unloading in the fruit, i.e., the very focus of the paper.

Because of the above considerations, I don’t think the paper provides information innovative enough to warrant its publication.

Following is a list of details (not all) that I noticed while studying it for this review.

Line 47-48: my understanding so far, but I may be wrong, was that TA is adverse to taste, as organic acids must be converted to sugars, or to pigments, aromas, etc. From this sentence, I gather that high TA is important for taste.

Line 50: what about structural carbohydrates? They are an important component of DMC, and textural attributes are related to them. You mention it later; I’d rather have it here.

Line 71: change ‘depend’ to ‘depends’.

Line 79: change ‘(DI) are increasingly’ should read ‘is increasingly’.

Line 86: ‘stress)’ misses a full stop.

Line 98: change ‘the 40th and 70th days after full bloom’ to ‘40 and 70 days after full bloom’.

Line 99: change ‘irritation’ to ‘irrigation’.

Line 107: change ‘is semi-arid’ to ‘are semi-arid’.

Line 112: ‘has not yet to be described’ needs reworking.

Fig. 1. Indicate on panel B the jdates of full bloom for the two seasons.

Line 128: the caption to figure 1 lists the data in the two panels wrongly.

Line 131-138: convert Imperial to metric units.

Line 131-138: experimental layout should be better illustrated. For example, were the 5 treatments applied to each block separately? If so, what about possible changes in soil characteristics? Also, how were 18 trees/block spatially arranged (e.g., were they on three adjacent rows? Else?). Finally, since 1.5 foot, or 45 cm spacing between trees is a very short distance, how far were the blocks from each other?

Line 154: change ‘schema’ to ‘scheme’.

Line 159: the abbreviation ‘degree d’ is not normally used. However, it should be defined the first time it is used.

Lines 162-164: this sentence makes very little sense in M&M. It is a discussion section comment and should be removed. Also, ‘degrees d’ is not correct English.

Line 173: 10 minutes is a very short time for water potential equilibration.

Line 175: cange 0using LICOR’ to ‘using a LICOR’.

Line 176: how many trees was Pn measured on? How many leaves per tree? Was care applied to randomize readings among treatments as time was passing?

Line 179-182. Just state that in 2019 wildfire smokes prevented the possibility to take Pn data. Incidentally, why were the fires limiting (since the LiCor was used with its light source)?

Lines 183-204: in reference to sections 2.3.2 and 2.3.3, where the measurements repeated on the same fruit during the season? Or where they randomly chosen? This should be provided, as it can impact the choice of ANOVA applied (repeated measures).

Line 215: please give reference for this formula.

Line 217: correct ‘IDA’.

Line 256: I assume Proc Mixed was used because of repeated measures? If so, I’d like to see it stated. If ‘normal’ ANOVA was instead justified, then proper indication of this should be given.

Lines 270-272: these are discussion more than result. This observation is only rendered here, but is applicable to the entire results section.

Lines 274-275: it is difficult for me to understand how DMC should decrease during summer, once DI was relieved, unless severe stresses were applied, like extremely high temps.

Table 2 and 3: lines after column AI have shifted one line down. Rearrange. Also, in footnote to table, I am not sure about the meaning of lettering. Since only lower case letters are used, it would seem that only comparisons within rows should be noted. In that case, however, the notation ‘within rows and years’ is not easy for me to understand. This applies to most tables in the paper.

Figure 2 and 3: please be consistent with treatment markers across paper figures.

Lines 305-306: there appears to be a leftover sentence here?

Line 306: please check whether year is only 2019 (I believe it’s not).

Line 318: how was fruit size derived from weight? I suppose using a conversion function. Was it developed in this study (then I think it should be described in M&M and some validation data presented), or derived elsewhere? A reference in the latter case would be nice.

Line 352: I suppose you mean ‘comparable’?

Line 373: in the footnote to table 5, I find the caption confusing. Once again it refers to ‘different letters within rows and years’. Within rows three separate parameters are listed, and the only difference is within columns, i.e., years. Please revise appropriately.

Line 403 and elsewhere: ‘Mpa’ should read ‘MPa’.

Line 480-484. Very old and not so surefire assessment of Pn.

Line 489: unable to relate reduction in cell turgidity (where are the data for this?) and stimulation of cell division. How late into the season does this last? I was unable to determine JD @ full bloom, therefore it is very hard to draw any conclusion whether during ED cell division might still be occurring.

Line 495-496: highly speculative and difficult to demonstrate. Were fruit shrivelled?

Lines 497-504. This discussion is highly speculative. Further, to address generically ‘mechanical process’ seems to allude to lack of knowledge of the vascular flows underpinning apple growth.

Line 506: your statement in this line is contradicted by your data and your own sentence two lines below. I don’t think correlations between DM and fruit quality can be so easily established. As you show, low water availability produced high DMC, but poor eating quality in DD. In particular, color formation is controlled by temp, as well as carbohydrates (they are needed for pigment synthesis).h

Line 508: by ‘in 2-yr’ you mean in the second year? Please use correct notation.

Line 511-512: I don’t understand this. Water is needed to ensure leaf functioning, hence photosynthesis. What does exactly mean “essential water supplement”?

Lines 512-514 Relationships between DMC and maturity levels are not to be sought: a fruit can mature either at low or high DMC. TA and firmness are measures of maturity, not DMC.

Line 547: this sentence doesn’t read correctly.

Line 562: remove description of Mr. Tayler. Simply indicate Tayler, personal communication. Since I doubt that Mr. Tayler could provide scientific data, a simple ‘anecdotical’ observation might be better, with no reference.

Lit. List

More than one style of citation appears to be implemented in this section.

Entry 22: missing initial for Serra

Entry 26: missing initial for Goodwin

Entry 29: please reformat Authors

Entry 36: this is not referenced in text

Reviewer 4 Report

Dear Author,

I have carefully reviewed your manuscript “Early summer deficit irrigation increases dry matter content and enhances quality of AmbrosiaTM apple at- and post-harvest”. The article promotes the adoption of deficit irrigation during early summer in order to increase dry matter content (DMC) and fruit quality of ‘Ambrosia’ apples. Five irrigation treatments are compared and their effects on DMC, soluble sugars, flesh firmness, colour, index of absorbance difference and soft scald are discussed.

Although the paper is relevant for both apple growers and the scientific community, I am afraid to say that this manuscript is not suitable for publication because contains serious flaws related to the structure, experimental design, scientific soundness and discussion.

The major flaws are reported below in the bullet points below. I have not added comments or edits to the rest of the manuscript, although needed, as the points below show enough evidence of rejection.

  • the version of the manuscript I have received was full of highlighted text and additions, probably derived from another review process. This did not help read through the paper, in which I found a lot of copy-and-paste content, repetitions of concepts and abbreviations. The English language used also needs moderate revision once the manuscript is substantially improved and modified.

  • The author makes the methodology difficult to follow by introducing 2 irrigation regimes at the start and then reporting 5 irrigation treatments, then mixing them up in a table. Readability would be much improved if the author only referred to 5 irrigation treatments (i.e., AI, ED, MD, LD and DD).

  • Stem water potential was measured on branches with sunlit leaves that were left equilibrating in enclosed bags (what kind of bags?) for only 10 minutes. Literature clearly reports that equilibrium is only achieved after more than 1 hour of enclosing leaves in bags. In addition the author does not report a critical information – i.e., the time of measurement. The physiological meaning of midday stem water potential is significantly different from pre-dawn water potential. The author likely only measured leaf water potential if bags were only used for ten minutes. Leaf water potential has a different meaning compared to stem water potential. This alone is a big bias in the experimental setup and these results cannot be discussed the way they are in this manuscript.

  • The author introduces the blush colour index (BCI) and an equation to calculate it but there is no reported rationale to do so. Literature shows that a* and hue angle are the CIELab / LCh colour attributes that best describe colour development against maturity or fruit quality. The author use L* to calculate BCI, but there is no reported evidence of the relevance of L* for this type of application.

  • Photosynthesis (Pn) was only measured in one season, and only at 2 times within the season, when 5 irrigation treatments with 4 different timings were used in this paper. The Pn results are therefore not relevant for the experimental design.

  • Flesh firmness is sometimes called FTA (fruit texture analyzer) interchangeably. This is wrong, the actual variable measured is flesh firmness and the unit used was Newton.

  • The means reported in Figures 2 and 3 are not separated using a post-hoc comparison (e.g., Fisher’s LSD or Tukey’s HSD) – so it is impossible to assess pairwise significant differences between treatments and over time.

  • Table 5 reports p-values of Year, Irrigation treatment, Stage and their interactions, but only reports means of the years 2008 and 2009. Why did the author not report means of stages, and irrigation treatments. Table 5 can probably be removed and results can be reported in the results section.

  • Figures 5 and 6 report an erroneous stem water potential (see comments above on how SWP was calculated). Change to leaf water potential.

  • Results reported in Table 6 are not introduced in materials and methods and are not relevant for the study, unless the author reports their rationale.

  • The author use Feet and Gallons as length and volume units, respectively. These need to be changed to SI units.

  • Not clear how the author measures DMC using the Felix F750 device. Was the generic apple model supplied by the manufacturer utilised in this study? In my experience these models are not good enough to predict DMC reliably in every cultivar. Therefore, additional training and validation on specific cultivars need to be used. Details of the model used and how it was developed (e.g., Partial Least Square regression or artificial neural network, any validation?) must be included.

I hope the comments will help improve the quality of upcoming research and to reconsider this paper’s publication without substantial changes to its structure and scientific soundness.

Kind Regards

Round 2

Reviewer 1 Report

The abstract does not reflect the content: description of irrigation treatments are so very detailed that half of the article is swept under Physiological investigation indicated...

A balanced and more adequate description of what was measured and what the results were should be given to all parts of the results.

Other comments:

Point 5: The author seems to have an interesting method of copying/pasting a comment from the reviewer into the text without bothering to look up references or acknowledging them.

I suggest uniform units in the author's response to point 6 L138-145:

The description of the water supplement has been revised to read, "Two irrigation regimes were implemented in the orchard: commercial irrigation (CI) and DI. Based on local industrial regime, CI was defined as 10 hours of irrigation occurred between 10 PM and 8 am for 10 days approximately, as 426 m3 / hectare per irrigation. Soil wet depth was 30 cm measured at the end of irrigation with 0.25 m3 / m3 of volumetric water content (VWC) (using 5TM soil sensors and was recorded at 30min interval using an EM50 data logger [25]). While DI treatment in this trial was designed as 4 hours of irrigation with the same irrigation system running for CI as 174 m3 / hectare per irrigation. The average wet depth was less than 15 cm after irrigation."

The wet zone, volumetric water content, and the amount of water added should be presented in a format that allows direct comparison. Here the author has provided information in three different units (four if we count L135). However, to avoid confusion, comparable units should be used, i.e. either meters or cm or dm.

L405 DI suppressed the value to -1.5 Mpa – the irrigation did not suppress the stem water potential, the latter simply decreased (or increased, depending on the reference point) due to insufficient water supply.

Reviewer 2 Report

All my concerns were addressed with success. My recommendation; Accept in present form.

Author Response

Thank you very much for your comments.